# The moderating and mediating role of eating behaviour traits in acceptance and commitment therapy-based weight management interventions: protocol for an individual participant data meta-analysis

Laura Kudlek ![ORCID],[1] Julia Mueller ![ORCID],[1] Patricia Eustacio Colombo,[1] Stephen J. Sharp,[1] Simon J. Griffin,[1,2] Amy Ahern ![ORCID] [1]

[1]MRC Epidemiology Unit, University of Cambridge, Cambridge, UK
[2]Primary Care Unit, University of Cambridge, Cambridge, UK

**Correspondence to**
Laura Kudlek;
laura.kudlek@mrc-epid.cam.ac.uk

## ABSTRACT

**Introduction** Precision medicine approaches to obesity aim to maximise treatment effectiveness by matching weight management interventions (WMIs) to characteristics of individuals, such as eating behaviour traits (EBTs). Acceptance and commitment therapy (ACT)-based WMIs may address EBTs such as emotional and uncontrolled eating more effectively than standard interventions, and might be most effective in people with high levels of these traits. However, few studies have examined this directly. We will examine (a) whether ACT-based interventions are more effective for people with certain levels of EBTs (ie, moderation) and (b) whether ACT-based interventions operate through changes in EBTs (ie, mediation).

**Methods and analysis** This individual participant data (IPD) meta-analysis will follow the Preferred Reporting Items for Systematic Reviews and Meta-Analyses of Individual Participant Data guidance. We will include studies on ACT-based WMIs that assessed EBTs in people with a body mass index ≥25 kg/m². We identified studies by screening studies included in a previous review of third wave cognitive behavioural interventions, and updating the search to 20 June 2022. We will request IPD from eligible published and unpublished studies. We will harmonise and re-analyse data using a two-stage random effects meta-analysis pooling within-trial interactions to investigate moderating effects and using a one-stage simultaneous equation model to examine mediating effects. We will assess the risk of bias in included studies using the Cochrane Risk of Bias tool 2 and the Risk of Bias in Non-randomised Studies of Interventions tool.

**Ethics and dissemination** Ethical approval has been obtained from the Cambridge Psychology Research Ethics Committee (Application No: PRE.2023.121). Data sharing will follow data transfer agreements and coauthorship will be offered to investigators contributing data. Findings will be disseminated through peer-reviewed journals and conferences and will contribute to the lead author's PhD thesis.

**PROSPERO registration number** CRD42022359691.

---

**STRENGTHS AND LIMITATIONS OF THIS STUDY**

⇒ Obtaining, harmonising and re-analysing individual participant data (IPD) from systematically identified studies allows us to investigate questions around mediation and moderation that individual studies are typically not powered to detect.

⇒ IPD might not be received from all eligible trials due to unresponsiveness, inability or unwillingness to share data and complex data sharing procedures. This might introduce a possibility for bias.

⇒ Individual studies are likely to exhibit heterogeneity in study populations, nature and intensity of interventions, duration of follow-up, outcome assessment and measures of eating behaviour traits, which may challenge the interpretation of data from pooled analyses and necessitate sensitivity analyses.

---

## INTRODUCTION

Obesity is associated with health risks, such as diabetes, metabolic diseases and several types of cancer.[1] Worldwide prevalence of overweight and obesity has increased almost threefold since 1975[1] and is now estimated to be around 60% in the UK[2] and 70% in the USA.[3] Behavioural weight loss treatments can achieve weight loss, leading to meaningful improvements in health, such as a reduced likelihood of developing diabetes.[4–6] However, outcomes of obesity treatments vary highly among individuals, with some individuals losing significantly more weight

than others, and only a proportion of people maintaining weight loss.[7] Precision medicine aims to understand what factors contribute to individual differences in treatment response, and how future intervention content and delivery can be adapted accordingly to maximise their net benefit.[8 9] Eating behaviour traits (EBTs) may be such a factor.[10–14] EBTs are defined as personal tendencies regarding an individual's reaction to food and food-related cues that determine the type, amount and frequency of food consumed, as well as when to start and stop eating.[15 16] Commonly measured EBTs are restraint (the conscious control of food intake to influence body weight), uncontrolled eating (the tendency to overeat in response to feelings of hunger and external stimuli) and emotional eating (the tendency to overeat in response to negative emotions).[17–19]

During behavioural weight management interventions (WMIs), restraint typically increases and uncontrolled and emotional eating decrease.[20–23] Standard behavioural WMIs involve dietary and physical activity advice, as well as behaviour change techniques such as self-monitoring, goal setting, planning, problem solving and cognitive restructuring.[24] However, individuals higher in emotional and uncontrolled eating might benefit from additional support targeting these EBTs.[10 12 25] Acceptance and commitment therapy (ACT) aims to increase psychological flexibility (ie, the capacity to remain in present moment awareness of one's thoughts, feelings, and sensations and accepting these) and to decrease experiential avoidance (ie, attempts to avoid unpleasant internal experiences).[26] WMIs based on ACT can therefore support the recognition of triggers of overeating as well as the acceptance of negative emotions and cravings, reducing the tendency to rely on food to relieve urges and regulate emotions. ACT-based WMIs are thus theorised to better address emotional and uncontrolled eating compared with standard behavioural techniques. A recent systematic review and meta-analysis concluded that ACT-based WMIs are effective at reducing emotional eating,[27] and a systematic review and network meta-analysis found ACT-based WMIs to be most effective at changing weight outcomes compared with other third wave cognitive behavioural therapies.[28]

However, to determine how useful EBTs and ACT-based WMIs could be for precision medicine approaches, it is also important to understand (a) whether ACT-based WMIs are more effective for people high in emotional and/or uncontrolled eating (ie, EBTs as effect moderators) and (b) whether changes in EBTs are mechanisms of effectiveness by which ACT-based WMIs impact weight outcomes (ie, EBTs as effect mediators). Although decreases in emotional and uncontrolled eating have been associated with improved weight loss in a variety of interventions,[12 21 29–31] few studies of ACT-based WMIs have examined the role of EBTs in detail. While studies of ACT-based WMIs often measure EBTs, they are mostly reported as outcomes, finding changes in EBTs in the desired direction.[32] Forman *et al* additionally explored the moderating role of EBTs, finding ACT-based interventions to be more effective than standard behavioural treatment in people with high initial levels of emotional eating in one study,[33] but not in another.[34] However, sample sizes of these studies were small (N=128 and N=190), and thus likely underpowered for the detection of moderating effects. To our knowledge, no studies of ACT-based interventions have yet examined mediating effects of changes in EBTs.

The scarcity of existing research that has directly examined the mediating and moderating role of EBTs prevents the synthesis of findings in a traditional meta-analysis and makes it difficult to form comprehensive conclusions. Individual participant data (IPD) meta-analysis allows us to collate and reanalyse raw data from all relevant studies that have measured the variables of interest, regardless of whether or not the studies have analysed or reported relationships of interest in the original publications.[35] This is likely to increase the number of included studies when compared with a traditional aggregate meta-analysis, and pooling IPD increases the power for analyses that individual studies alone may not be powered to detect. Additionally, heterogeneity is reduced by harmonising data from different measures and re-analysing IPD from original studies according to a uniform analysis plan.[35] Using an IPD meta-analysis thus increases the quantity and quality of data on EBTs in ACT-based WMIs beyond existing evidence, generating a resource to address questions around the moderating and mediating role of EBTs in ACT-based WMIs. This will provide us with new understanding about the potential to use ACT-based interventions to support people with high levels of EBTs as part of a precision medicine approach to obesity treatment that directs specific interventions to those that are most likely to benefit from them.

## Objectives

The aim of this IPD meta-analysis is to obtain and re-analyse IPD from studies assessing EBTs in ACT-based WMIs to examine:

1. To what extent the effect of ACT-based WMIs on weight loss depends on individuals' levels of EBTs.
2. To what extent changes in EBTs mediate the effect of ACT-based WMIs on weight loss.

## METHODS AND ANALYSIS

This project will follow guidance from the Preferred Reporting Items for Systematic Reviews and Meta-Analyses for Individual Participant Data (PRISMA-IPD) extension.[36] The protocol will additionally follow the Preferred Reporting Items for Systematic Review and Meta-Analysis Protocols (PRISMA-P) where applicable. The phases of planning and conducting an IPD meta-analysis differ from those of standard aggregate systematic reviews and meta-analyses, with some occurring concurrently.[37] We will thus communicate the current stage of work on the project by using either past or future tense to describe relevant sections.

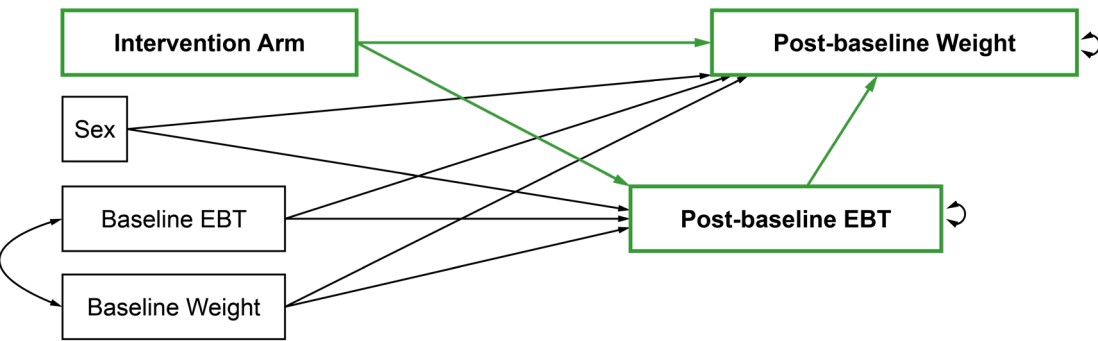

**Figure 1** Path diagram for mediation analysis of individual eating behaviour traits (EBTs).

## Study identification
### Eligibility criteria

Data from published and unpublished studies were considered eligible for inclusion if studies met the following criteria:

1. Population: adults (aged 18 and older) with a body mass index (BMI) ≥25 kg/m². Studies were excluded if participants were recruited purely based on having a chronic disease or being pregnant, as were studies where eligible participants resided in institutional settings (eg, hospital, army barracks). Studies on children and adolescents were not considered for inclusion to avoid the risk of increasing heterogeneity in interventions serving different target populations.
2. Intervention: interventions with the primary goal of weight loss or weight loss maintenance (referred to as WMIs in the remainder of this protocol) that report incorporating strategies based on ACT. ACT-based interventions from different contexts were eligible (eg, online, in person, healthcare setting, commercial), and they were eligible either as standalone treatment or as part of a wider WMI.
3. Comparators: no/wait-list control, minimal intervention (eg, leaflet, brief advice) or standard behavioural WMI. Studies with no control group will be eligible for a subset of analyses.
4. Outcome: weight assessed at post-treatment or both at post-treatment and any follow-up point. A follow-up point of at least 3 months post-baseline had to be available.
5. Mediators/moderators: EBTs assessed at baseline, post-treatment or both. Eligible EBTs are emotional eating, uncontrolled eating, disinhibition, external eating and restraint. Disordered eating will be excluded.
6. Study design: due to the scarcity of research in the field, we will include all types of intervention studies to obtain as much IPD as possible (randomised controlled trials (RCTs), non-RCTs, prospective cohort and case series studies). However, only RCTs and cluster-RCTs will be eligible for the full set of analyses.

### Search strategy

We screened studies included in a review by Lawlor *et al*[28] on third wave cognitive behavioural therapies for weight management against this review's eligibility criteria. In addition, we re-ran an adapted search relating to the concepts of (a) ACT and (b) overweight, obesity or weight management from 25 September 2019 until 20 June 2022 (see online supplemental material). Concepts relating to other third wave cognitive behavioural therapies were deleted from the original search strategy to match the narrower focus of this IPD meta-analysis. No restrictions on language were applied. We searched the databases MEDLINE, CINAHL, Embase, PsycINFO, AMED, ASSIA, Web of Science, the Cochrane database (CENTRAL) and hand-searched the reference lists of key publications. Furthermore, when requesting IPD, all authors of included studies will be asked whether they are aware of any additional eligible published or unpublished data.

### Study selection

Two independent reviewers, PEC and LK, screened both title and abstracts and full texts in Covidence.[38] Disagreements were resolved by discussion and consulting of a third reviewer, AA, where necessary. We will contact authors of protocols and conference abstracts to resolve any outstanding questions on eligibility and to enquire about their willingness to share unpublished results.

### Risk of bias assessment

The Cochrane Risk of Bias tool 2 (RoB2)[39] or the Risk of Bias in Non-randomised Studies of Interventions tool (ROBINS-I)[40] will be used by two researchers independently to assess risk of bias. Disagreements will be resolved by discussion and consulting of a third reviewer if necessary.

### Variables requested

The following variables will be requested in a detailed data dictionary, including definitions:

1. De-identified participant identification number.
2. Age at baseline.
3. Sex.
4. Height at baseline.
5. Weight at baseline, end of intervention and any follow-up.
6. Allocated trial arm.
7. Number of sessions attended.

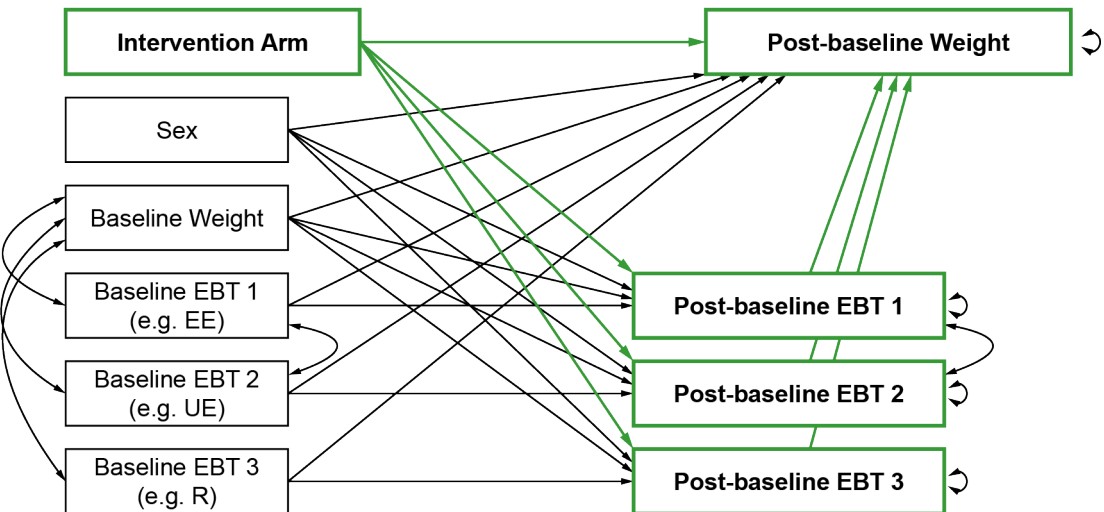

**Figure 2** Joint path diagram for mediation analysis combining individual eating behaviour traits (EBTs) (eg, emotional eating (EE), uncontrolled eating (UE) and restraint (R)).

8. All measured EBTs at baseline, end of intervention and any follow-up point.
9. All measures of experiential avoidance and/or psychological flexibility at baseline, end of intervention and any follow-up point.
10. Variables that describe which participants were excluded from main analysis and why.

### Data collection and management

#### Extraction of published data on study characteristics

Data on study characteristics will be extracted from published manuscripts by two reviewers independently, using a form that will be adapted from the Cochrane data extraction form (online supplemental material).[41] These include data on study design and setting, participant sociodemographic characteristics (eg, ethnicity, education, socioeconomic status) and details on the intervention and control conditions. A third reviewer will cross-check extractions for any discrepancies, and original study authors will be provided with an opportunity to cross-check extractions for accuracy, including any potential updates.

#### Requesting IPD

Authors will be contacted via email, outlining the purpose of the study, providing the study protocol and asking them to collaborate on the project by providing IPD. If the corresponding authors agree, more detailed instructions including an outline of the specific data requested and a data dictionary will be shared. If authors are not able to share their data, we will ask them to perform the analyses using a prespecified protocol and share the results so that we can include them in the meta-analysis. Authors will be sent two reminders after initial contact, each with a time period of around 3 weeks between them. If no response is given and published results are not suitable for meta-analysis, the study concerned will be excluded.

Studies that are excluded due to inability to contact will be summarised in the final publication.

#### Collecting IPD

Data will be accepted in any format, but a Microsoft Excel format is preferred. A data dictionary will be provided prior to data transfer. Any data sharing will strictly follow the conditions prespecified in data transfer agreements, and all data will be shared via, and stored in the MRC Epidemiology Unit's secure research drive. After receiving data, a copy will be saved that will be kept as original. Working files will be converted and imported into R V.4.1.2.[42]

#### Data harmonisation

Data will be harmonised according to the prespecified data dictionary to merge it into a combined dataset. Variables will be recoded and transformed using a uniform coding scheme (eg, height and weight will be transformed into metric units, age will be converted into age in years, etc). EBTs are likely to have been assessed using different questionnaires across studies (eg, TFEQ-R18, TFEQ-51, DEBQ, etc). To harmonise EBT data, we will ensure that all EBT outcome scores represent the relative proportion of highest possible raw scores on a 0–100 range. To convert EBT outcome scores that do not follow this scoring approach, we will subtract the raw outcome score by the lowest possible raw score, divide this by the possible score range and multiply this by 100. To facilitate this, we will use item-level data where available.

#### Data checking

Data received will be compared with that reported in the original publication by running descriptive statistics (eg, sample size, weight loss outcomes). Additionally, if both item and subscale levels of EBTs are provided, subscales will be re-computed and compared with those provided

to confirm item scoring and ensure the uniform allocation of items to subscales. In case of any discrepancies, authors will be contacted for clarification. If discrepancies cannot be resolved, the nature and severity of the deviation will be evaluated to determine whether the study can be included. After a study has been checked, a summary will be sent to the original authors to give them the opportunity to address any issues before merging the study into the pooled dataset.

### Database creation and aggregation

A merged dataset containing harmonised data from all included studies will be created for convenience when managing and analysing data. Data from different studies will be distinguished using a unique identifier for each study that is allocated before the final merging. To check for any errors introduced during the merging phase, descriptive statistics of each study will be run before merging datasets and compared with corresponding statistics produced after merging.

### Studies where IPD is not available

If authors are unable to share their raw data, we will ask them to perform the required analyses for us and share their results (ie, a federated approach). This will allow us to still include the study in the moderation analyses. To check whether this might introduce bias, we will conduct sensitivity analyses using only the data from studies providing IPD. If authors can neither provide IPD, nor provide the outcome statistics of requested analyses (ie, federated data), any published data will be synthesised and compared with included studies. This will include available published data on sociodemographic variables, such as socioeconomic status, sex and age, as well as any reported data on eating behaviours at baseline and end of intervention, and weight data at baseline, end of intervention and follow-up. As with data on study characteristics, published data on EBTs and weight outcomes will be extracted in duplicate using a prespecified data extraction form based on the Cochrane data extraction form template.[41]

### Data analyses
#### Descriptive statistics
Descriptive statistics and the proportion of missing values of age, sex, baseline EBTs and baseline BMI will be derived from the raw data directly. These will be reported for each study, as well as for the overall sample included in the meta-analyses. Other study characteristics, of which IPD will not be requested, for example, socioeconomic status and ethnicity, will be extracted from published reports.

#### Statistical analysis
To gain a more complete picture of the role of EBTs in WMIs, we will explore to what extent (a) baseline EBTs are associated with changes in weight over the intervention and follow-up period and (b) to what extent changes in EBTs are associated with changes in weight over the intervention and follow-up period before conducting the main moderation and mediation analyses. These initial associations will be explored using linear regression, adjusted for study arm, age, sex, baseline weight and duration of follow-up. This will allow us to incorporate evidence from studies without a control group. Findings will be synthesised using a two-stage IPD meta-analysis approach,[35] in which analyses are first performed for each study individually, and then the results are combined using meta-analysis. This allows for the incorporation of studies that do not provide IPD if collaborators perform requested analyses at their home institutions and share only their output. We will use random-effects meta-analyses to accommodate for heterogeneity between studies.

#### Moderation
To investigate whether the effect of ACT-based interventions on weight depends on people's levels of emotional and uncontrolled eating, we will use a two-stage IPD meta-analysis approach and perform a random-effects meta-analysis to combine within-trial regression coefficients of the intervention*EBT interactions. To facilitate interpretation of potential interaction effects with EBT as a continuous variable, we will also fit regression models in three EBT subgroups (reflecting 'low', 'medium' and 'high' levels of the respective EBT) in each trial and combine the estimated subgroup-specific effects using random-effects meta-analysis. Subgroups will be determined by using the tertile EBT scores of the pooled sample as cut-offs for each individual trial. We will investigate the potential moderating effects of EBTs on weight loss outcomes at the end of the intervention as well as at available follow-up timepoints (eg, 6 and 12 months).

#### Mediation
Mediation analyses will follow 'A Guideline for Reporting Mediation Analyses' (AGReMA) guidelines.[43] We will use the simultaneous equations model (SEM) approach outlined by Huh et al. (2022)[44] to estimate (a) the direct effect by which ACT-based interventions impact weight loss, (b) the indirect effect by which they impact weight loss via changes in EBTs and (c) the total effect by which weight loss is impacted. We will estimate direct, indirect and total effects on weight loss outcomes at the end of the intervention as well as at available follow-up timepoints (eg, 6 and 12 months). Figures 1 and 2 depict the path diagrams that will be tested both in each individual trial ('study-specific sub-models') and in a pooled sample using a one-stage approach ('overall model'). The SEM for the overall models will control for clustering within studies by using complex survey analysis. While the one-stage approach eliminates the risk of aggregation/ecological bias, it requires IPD, making a federated approach (where authors share only their output) impossible. Thus, studies that cannot provide IPD will be excluded for this part of the analysis. If trials with less than 50 participants per intervention arm will be included, we will consider adapting the path diagram depicted in figures 1 and 2 by adding the variables age and height with a path to

both the outcome and intervention arm to control for possible imbalances between the randomised groups. If it is concluded from previous moderation analyses that ACT-based interventions only have a significant effect on weight outcomes in one or two of the three EBT level subgroups (low, medium or high EBT levels), then we will repeat the mediation analyses in the subgroup(s) that did show a significant effect on weight outcomes. Analyses will be performed in R,[42] using the lavaan[45] and lavaan. survey[46] packages.

### Sensitivity analyses

We will conduct a series of sensitivity analyses. At the study level, we will compare results from analyses performed in (a) the full set of studies versus excluding studies classified at high risk of bias using the RoB2 (also see the Risk of bias assessment section); (b) studies providing IPD versus studies providing federated data only (also see the Studies where IPD is not available section); (c) studies with waitlist or minimal control conditions versus studies with standard behavioural control conditions; (d) studies that significantly reduced experiential avoidance (as this is the hypothesised mechanism of action of ACT-based WMIs) versus studies that did not significantly reduce experiential avoidance. At the individual level, we will compare results from analyses performed in; (e) all individuals versus those that received a sufficient dose of the intervention (as determined from liaising with study authors). Details will be described in a separate analysis plan that will be shared via the Open Science Framework.[47]

### Patient and public involvement

We will seek patient and public involvement (PPI) input for the interpretation of analyses and implication of results in the form of remote focus group meetings with two or more PPI members. This may result in coauthorship on the publication.

### ETHICS AND DISSEMINATION

Ethical approval was obtained from the Cambridge Psychology Research Ethics Committee (Application No: PRE.2023.121) on 24/11/2023. Data from individual trials will be shared with the lead institution under appropriate data sharing agreements. We will disseminate findings through peer-reviewed journals and conferences. Investigators contributing data to this analysis will be offered coauthorship. This study will contribute to the lead author's PhD thesis.

**Contributors** LK conceived and designed study plans, developed the analysis strategy, performed initial screening and drafted the protocol manuscript. AA provided input on the conception and development of study plans, including the analysis strategy, contributed to screening and reviewed drafts of the manuscript. JM contributed to the development of study plans, in particular the analysis strategy, and reviewed drafts of the manuscript. SS provided input on the analysis strategy and reviewed drafts of the manuscript. PEC provided input on study design, performed initial screening and reviewed drafts of the manuscript. SJG provided input on study design and reviewed drafts of the manuscript. All authors have read and approved the final version of the manuscript.

**Funding** This work was supported by the Medical Research Council MC_UU_00006/6 as part of LK's PhD.

**Competing interests** AA, JM, SS and SJG were authors of two studies identified as eligible for this IPD meta-analysis (SWiM-F and SWiM-C). JM is a Trustee for the Association for the Study of Obesity (unpaid role). SJG has received honoraria from Astra Zeneca for contributing to postgraduate education sessions for primary care teams concerning the management of type 2 diabetes. SJG is a trustee of the Novo Nordisk UK Research Foundation. AA is a member of the WW Scientific Advisory Board.

**Patient and public involvement** Patients and/or the public were involved in the design, or conduct, or reporting, or dissemination plans of this research. Refer to the Methods section for further details.

**Patient consent for publication** Not applicable.

**Provenance and peer review** Not commissioned; externally peer reviewed.

**ORCID iDs**
Laura Kudlek http://orcid.org/0000-0003-0947-3640
Julia Mueller http://orcid.org/0000-0002-4939-7112
Amy Ahern http://orcid.org/0000-0001-5069-4758

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
