## [Reviewer comments · BMJ Open]

ARTICLE DETAILS

TITLE (PROVISIONAL)	The Moderating and Mediating Role of Eating Behaviour Traits in Acceptance and Commitment Therapy-based Weight Management Interventions: Protocol for an Individual Participant Data Meta-analysis
AUTHORS	Kudlek, Laura; Mueller, Julia; Eustacio Colombo, Patricia; Sharp, Stephen; Griffin, Simon; Ahern, Amy

VERSION 1 – REVIEW

REVIEWER	Hinton, Elanor University of Bristol
REVIEW RETURNED	07-Aug-2023

GENERAL COMMENTS	This proposed systematic review and meta-analysis aims to examine whether weight management interventions are moderated and/ or mediated by eating behaviour traits. From my understanding of the literature, this is a novel proposal to use individual participant data meta-analysis, follows the appropriate PRISMA guidelines, and will add value and understanding to the field. The protocol is well written and the authors have considered the involvement of authors of the included papers appropriately (given the IPD analysis). I have a few minor comments for clarification prior to acceptance of this paper, as follows: Pg 2, Line no. approx26/27: As this is a protocol paper, it isn't clear why the search has been limited up to 20.06.2022. Please can the authors clarify this, and also whether it is the Lawlor et al (2020) search that was updated. Will the search be updated upon acceptance of the protocol for publication? Section 3 Introduction: to widen the understanding and value of this manuscript to a larger audience, I suggest a brief introduction to ACT itself would be helpful (in addition to the explanation around ACT-based WMIs for recognition of food cues etc). This should also include brief explanation of terms such as experiential avoidance and psychological flexibility used later in section 4.3 and section 4.2.5.3 (where it is noted that experiential avoidance is the hypothesised mechanism of action of ACT-based WMIs – I think this should be introduced upfront for greater clarity). Section 3.1 Objectives: it is not clear from the background provided in the introduction why restraint is not included as one of the potential moderating EBTs but is included as a potential mediator. A brief note to clarify this issue should be provided in the introduction.
---

	Section 4: Somewhat unusually perhaps, it appears some of this work has already been completed (e.g. literature search following Lawlor et al), prior to submission and acceptance of this protocol paper. If that is acceptable to the journal (procedures do appear sound and thorough), this point should be made explicit early on, in addition to the nod to ongoing work in the methods section. It's not clear why the search (conducted over a year ago in June 2022) was not conducted closer to submission of the protocol paper – i.e. in 2023 to capture the most up to date literature in the field. Can the authors clarify? It is possible that any additional papers published between June 2022 and now may be captured by asking authors of the included studies for additional papers but it should be noted that this is unlikely to be as systematic as re-running the searches. Section 4.1.1: – please can the authors clarify the justification for including only adults and not young people in the review. – mediators and moderators assessed at inclusion – were any EBTs included or just the three (restraint, uncontrolled and emotional eating) mentioned in the introduction? Please clarify in the text. Section 4.4.1: please can the authors provide the draft data extraction form in an appendix (as well as the ref to Cochrane template). Section 4.4.4: please can the authors provide further details of how the EBT values will be standardised across questionnaires. NB for section 4.5.2.1 – I have not commented on this section of the proposed analysis as do not have specific expertise in IPD MA. Section 4.2.5.3 is missing the 4 in the heading (page 11), I think. Also the authors mention data regarding 'a sufficient dose of the intervention' – this should be added to the section on variables requested (section 4.3), perhaps relating to the number of sessions attended (but presumably additional information from original study authors regarding what was deemed 'sufficient'). Do the authors plan to share the separate analysis plan and if so where (OSF perhaps?).
--	--

REVIEWER	Embling, Rochelle Swansea University, School of Psychology
REVIEW RETURNED	21-Aug-2023

GENERAL COMMENTS	Thank you for inviting me to review this protocol. The topic is well-justified and details of the planned review and meta-analysis are thoroughly explained. A few minor comments are included for further clarification below if helpful: Line 13 – 16: Will this include studies where participants are referred to interventions as part of treatment pathways? Might be worth clarifying in addition to describing as community dwelling. Line 27: Potential typo – post-treatment stated twice? Does this mean post-treatment only, or post-treatment AND any follow-up point? Will this measure pre-post change?
---

	Line 48: Searches may be re-run to further update included data as of 2022-2023. Line 58-59: Re grey literature: In addition to contacting authors of included studies, will other grey literature also be sought? E.g., Grey literature databases (OpenGrey), relevant conference proceedings, thesis data. Can see this is briefly mentioned in Study Selection. Line 44 – 48: Just a thought that scoring across and within questionnaires may also differ. Eg TFEQ subscales have different numbers of items and there are also many questionnaire versions that adapt response options. Might be worth clarifying how values will be standardised – e.g., convert to proportional scores (so all 0 – 100 scores?). 4.5.2 statistical analysis – Might be worth noting reason for random effects models (e.g., expected heterogeneity as mentioned previously). 4.5.2.1 & .2 Moderation and mediation - I understand that it may difficult to calculate sample size needed, but given the complexity of pathway models, might be worth commenting on the expected sufficiency of data across studies in combined datasets to enable these (given that as a rough rule of thumb, sample size would typically need to be well into the hundreds for individual datapoints). Equally, you might consider the use of bootstrapping to determine significance, and define what you would consider to be a 'meaningful' effect/ change in the outcome before running these analyses. 5. Ethics and dissemination – I would also mention here preregistration of the study protocol in Prospero, in line with guidelines for health-focussed systematic reviews and meta-analyses (if not mentioned elsewhere in main text). Might also be worth clarifying if the resulting homogenised dataset for this meta-analysis will be made open access (to support future updates to the review/ meta-analysis in line with Cochrane recommendations)
--	--

VERSION 1 – AUTHOR RESPONSE

Reviewer #1		
Overall comments:		
This proposed systematic review and meta-analysis aims to examine whether weight management interventions are moderated and/ or mediated by eating behaviour traits. From my understanding of the literature, this is a novel proposal to use individual participant data meta-analysis, follows	Thank you for the review of our manuscript and the comments below.	NA

the appropriate PRISMA guidelines, and will add value and understanding to the field. The protocol is well written and the authors have considered the involvement of authors of the included papers appropriately (given the IPD analysis). I have a few minor comments for clarification prior to acceptance of this paper, as follows:		
Minor comments: 1. Pg 2, Line no. approx26/27: As this is a protocol paper, it isn't clear why the search has been limited up to 20.06.2022. Please can the authors clarify this, and also whether it is the Lawlor et al (2020) search that was updated. Will the search be updated upon acceptance of the protocol for publication?	Thank you so much for this comment. Also see comment 4. Planning and organisation of this IPD-meta-analysis (including when to conduct searches etc.) was guided by a book on IPD meta-analyses by Riley et al. (2021).² Time management around conducting searches for included studies for IPD meta-analyses is typically very distinct from standard aggregate systematic reviews and meta-analyses. Since an IPD meta-analyses requires sharing of raw data and setting up data sharing contracts with the legal teams of the respective universities, it takes a long time to obtain the requested data (typically one to two years should be calculated for collecting IPD). As a consequence, "the protocol will usually be completed after the searches for trials and subsequent screening have been completed".³ Given the long time it takes to collect IPD data and the typical proceedings of IPD meta-analyses, we will not update the search upon acceptance of the protocol.	NA
2. Section 3 Introduction: to widen the understanding and value of this manuscript to a larger audience, I suggest a brief introduction to ACT itself would be helpful (in addition to the explanation around ACT-based WMIs for recognition of food cues etc). This should also include brief explanation of terms such as experiential avoidance and psychological flexibility used later in section 4.3 and section 4.2.5.3 (where it is noted that experiential avoidance is the hypothesised mechanism of action of ACT-based WMIs – I think this	Thank you for this comment. I have added a more general statement about ACT itself to the introduction and included definitions of psychological flexibility and experiential avoidance. The final publication of results of this project will include more detail on ACT theory and its components. "Acceptance and commitment therapy (ACT) aims to increase psychological flexibility (i.e. the capacity to remain in present moment awareness of ones thoughts, feelings, and sensations and accepting these) and to decrease experiential avoidance (i.e. attempts to avoid unpleasant internal experiences). 27 WMIs based on	p. 4, lines 35 to 50

should be introduced upfront for greater clarity).	ACT can therefore support the recognition of triggers of overeating as well as the acceptance of negative emotions and cravings, reducing the tendency to rely on food to relieve urges and regulate emotions. ACT-based WMIs are thus theorised to better address emotional and uncontrolled eating compared with standard behavioural techniques.”	
3. Section 3.1 Objectives: it is not clear from the background provided in the introduction why restraint is not included as one of the potential moderating EBTs but is included as a potential mediator. A brief note to clarify this issue should be provided in the introduction.	Thank you so much for spotting this. All eligible EBTs should be examined as both moderators and mediators, so we have updated the research questions to reflect this.: “(a) to what extent the effect of ACT-based WMIs on weight loss depends on individuals’ levels of EBTs (b) to what extent changes in EBTs mediate the effect of ACT-based WMIs on weight loss”	p. 5, lines 44 to 48
4. Section 4: Somewhat unusually perhaps, it appears some of this work has already been completed (e.g. literature search following Lawlor et al), prior to submission and acceptance of this protocol paper. If that is acceptable to the journal (procedures do appear sound and thorough), this point should be made explicit early on, in addition to the nod to ongoing work in the methods section. It’s not clear why the search (conducted over a year ago in June 2022) was not conducted closer to submission of the protocol paper – i.e. in 2023 to capture the most up to date literature in the field. Can the authors clarify? It is possible that any additional papers published between June 2022 and now may be captured by asking authors of the included studies for additional papers but it should be noted that this is unlikely to be as systematic as re-running the searches.	Thank you so much for this comment. Planning and organisation of this IPD-meta-analysis (including when to conduct searches etc.) was guided by a book on IPD meta-analyses by Riley et al. (2021).² Please see comment 1 for details. However, to help capture up-to-date literature, we have asked authors of study protocols and trial registries to share IPD from unpublished trials. To clarify that both published and unpublished data was eligible, I have added some wording on page 6 (section “4.4.1 eligibility criteria”): “Data from published and unpublished studies were considered eligible for inclusion if studies met the following criteria”	p. 6, lines 14 to 16
5. Section 4.1.1: please can the authors clarify the justification for including only adults and not young people in the review.	We focussed on adults for various reasons. Mainly, we aimed to keep a narrow scope for this project. This is due to the complex nature of IPD meta-analyses and the amount of	p. 6, lines 23 to 24

	time, planning and administrative capacity required to complete such a project. Additionally, interventions might differ for adults and young people, which would pose the risk of them being to heterogenous to combine in one single analysis. „Studies on children and adolescents were not considered for inclusion to avoid the risk of increasing heterogeneity in interventions serving different target populations.”	
6. Section 4.1.1: mediators and moderators assessed at inclusion – were any EBTs included or just the three (restraint, uncontrolled and emotional eating) mentioned in the introduction? Please clarify in the text.	Thank you so much for this comment. We have updated the eligibility to clarify. We focussed on the following EBTs: Restraint, uncontrolled eating, disinhibition, external eating, emotional eating. We chose to focus on these EBTs as they are most commonly assessed and have been linked to obesity in previous literature. “Mediators/ Moderators: EBTs assessed at baseline, post-treatment or both. Eligible EBTs are emotional eating, uncontrolled eating, disinhibition, external eating and restraint. Disordered eating will be excluded.”	p. 6, lines 39 to 42
7. Section 4.4.1: please can the authors provide the draft data extraction form in an appendix (as well as the ref to Cochrane template).	The reference to the Cochrane template is provided on page 14 (ref no 43). We have added the data extraction form to the supplementary materials	p. 16, lines 7 & 8 + Supplementary Material
8. Section 4.4.4: please can the authors provide further details of how the EBT values will be standardised across questionnaires.	We have added more detail to the manuscript. We will be bringing EBT scores to a range from 0 to 100 using item-level data. If item level data is not provided, we will use raw subscale scores. If these are also not provided, and the outcome is not already on a 0 to 100 scale, then we will have to consider alternative methods of harmonization or excluding the study in question. “To harmonise EBT data, we will ensure that all EBT outcome scores represent the relative proportion of highest possible raw scores on a 0 to 100 range. To convert EBT outcome scores that do not follow this scoring approach, we will subtract the raw outcome score by the lowest possible raw score, divide this by the possible score range and multiply this by 100.”	p. 8 line 59 to page 9 lines 3 to 8

	To facilitate this, we will use item-level data where available.	
9. Section 4.2.5.3 is missing the 4 in the heading (page 11), I think. Also the authors mention data regarding ‘a sufficient dose of the intervention’ – this should be added to the section on variables requested (section 4.3), perhaps relating to the number of sessions attended (but presumably additional information from original study authors regarding what was deemed ‘sufficient’). Do the authors plan to share the separate analysis plan and if so where (OSF perhaps?).	The section heading has been corrected. The number of sessions attended has been reported in the variables requested section. What dose of the intervention is considered sufficient will not be collected in the form of a variable, but instead will be discussed with collaborators via email. Depending on their responses, and the data received, we will either agree on a percentage of number of sessions that will be applied across studies to categorise sufficient attendance, or we will consider percentage of attendance on a case-by-case basis. This can only be decided after authors replies, since we do not know at this stage if all authors will be able to provide us with insights on what they considered to be sufficient attendance or not. Thank you for this comment, we agree it would be a good idea to share the analysis plan once finalised. We will share the analysis plan on the OSF and have adapted the protocol accordingly on page 11: “Details will be described in a separate analysis plan that will be shared via the Open Science Framework (OSF).”	p. 11, line 54
Reviewer #2		
Overall comments: Thank you for inviting me to review this protocol. The topic is well-justified and details of the planned review and meta-analysis are thoroughly explained. A few minor comments are included for further clarification below if helpful:	Thank you for the review of our manuscript and the comments below.	NA
Minor comments: 1. Line 13 – 16: Will this include studies where participants are referred to interventions as part of treatment pathways? Might be worth clarifying in addition to describing as community dwelling.	Yes, they will. With “community dwelling” we were referring to participants living independently. Basically, studies that provide interventions to participants who live at the treatment site or who all live in an institutional context (e.g. hospital, nursing home, army), were excluded. We have changed the wording to clarify this, and we have dropped the	p. 6, lines 21 to 22

	term “community dwelling” as it might cause more confusion than clarity. “Population: Adults (aged 18 and older) with a BMI ≥ 25 kg/m2. Studies were excluded if participants were recruited purely based on having a chronic disease or being pregnant, as were studies where eligible participants resided in institutional settings (e.g. hospital, army barracks)”	
2. Line 27: Potential typo – post-treatment stated twice? Does this mean post-treatment only, or post-treatment AND any follow-up point? Will this measure pre-post change?	Thank you for spotting this. We have added a few words to clarify. “Outcome: Weight assessed at post-treatment or both at post-treatment and any follow-up point. A follow-up point of at least 3-months post-baseline had to be available.”	p. 6, lines 36 to 37
3. Line 48: Searches may be re-run to further update included data as of 2022-2023.	Planning and organisation of this IPD-meta-analysis (including when to conduct searches etc.) was guided by a book on IPD meta-analyses by Riley et al. (2021).² Time management around conducting searches for included studies for IPD meta-analyses is typically very distinct from standard aggregate systematic reviews and meta-analyses. Since an IPD meta-analyses requires sharing of raw data and setting up data sharing contracts with the legal teams of the respective universities, it takes a long time to obtain the requested data (typically one to two years should be calculated for collecting IPD). Given the long time it takes to collect IPD data and the typical proceedings of IPD meta-analyses, we will not be able to update the search. This is common for IPD meta-analyses. However, to help capture up-to-date literature, we have asked authors of study protocols and trial registries to share IPD from unpublished trials. To clarify that both published and unpublished data was eligible, I have added half a sentence on page 6 (section “4.4.1 eligibility criteria”) “Data from published and unpublished studies was considered eligible for inclusion if studies met the following criteria”	p. 6, lines 14 to 16
4. Line 58-59: Re grey literature: In addition to contacting authors of included studies, will other grey literature also be sought?	No other grey literature databases were searched. However, these trial registries and conference abstracts, were eligible for inclusion. Additionally,	NA

E.g., Grey literature databases (OpenGrey), relevant conference proceedings, thesis data. Can see this is briefly mentioned in Study Selection.	we considered both completed and ongoing studies as eligible, so authors of protocols or trial registries were also contacted to share data.	
5. Line 44 – 48: Just a thought that scoring across and within questionnaires may also differ. Eg TFEQ subscales have different numbers of items and there are also many questionnaire versions that adapt response options. Might be worth clarifying how values will be standardised – e.g., convert to proportional scores (so all 0 – 100 scores?).	That is correct. Since we will have item-level data available in hopefully most cases, we will re-calculate or convert outcomes to ensure they are on a range from 0 to 100 (as the TFEQ scoring already is if scored correctly). If item level data is not provided, we will use raw subscale scores. If these are also not provided, and the outcome is not already on a 0 to 100 scale, then we will have to consider alternative methods of harmonization or excluding the study in question. We have added more detail to the manuscript to clarify. “To harmonise EBT data, we will ensure that all EBT outcome scores represent the relative proportion of highest possible raw scores on a 0 to 100 range. To convert EBT outcome scores that do not follow this scoring approach, we will subtract the raw outcome score by the lowest possible raw score, divide this by the possible score range and multiply this by 100. To facilitate this, we will use item-level data where available.”	p. 8, line 60
6. 4.5.2 statistical analysis – Might be worth noting reason for random effects models (e.g., expected heterogeneity as mentioned previously).	We have added a note of this in the manuscript: “We will use random-effects meta-analyses to accommodate for heterogeneity between studies.”	p. 10, line 34
7. 4.5.2.1 & .2 Moderation and mediation - I understand that it may difficult to calculate sample size needed, but given the complexity of pathway models, might be worth commenting on the expected sufficiency of data across studies in combined datasets to enable these (given that as a rough rule of thumb, sample size would typically need to be well into the hundreds for individual datapoints).	Thank you for your comment. As you said, mediation analyses will usually require a sample size of several hundreds. Thus, single trials are usually not powered to do mediation analyses, and these research questions remain unaddressed. That is why this IPD meta-analysis provides such a unique opportunity where we can pool data from several trials to examine this research question with more power. To highlight the power advantage over individual studies, we have added wording to the last paragraph of the introduction: “Individual Participant Data (IPD) meta-analysis allows us to collate and reanalyse raw data from all relevant	p. 5, lines 23 to 24

	studies that have measured the variables of interest, regardless of whether or not the studies have analysed or reported relationships of interest in the original publications. This is likely to increase the number of included studies when compared to a traditional aggregate meta-analysis, and pooling IPD increases the power for analyses that individual studies alone may not be powered to detect.” This is also described in the first bullet point of the strengths and limitations on page 3. To our knowledge, there are several more specific sample size recommendations for mediation analyses available in the literature. For example, some rules of thumb recommend a ratio of 5:1 or 10:1 observations to free parameters. Since the number of EBTs included in the path diagrams and hence the number of free parameters will depend on the data we receive, we did not apply such rules in this protocol. Other rules of thumb revolve around specific numbers, such as 500. However, we found them to often disagree, without clear guidance on how they are derived and which to choose. As with any meta-analysis, we will include all eligible data and that was systematically identified. This should be more than 500 if we include more than approximately 6 trials, which is highly likely. Since we will not stop “recruiting” at a specific sample size target, as might be the case for primary data collection, we did not determine a specific sample size target for the protocol beyond the discussion of increased power due to pooling data from several studies.	
8. Equally, you might consider the use of bootstrapping to determine significance, and define what you would consider to be a ‘meaningful’ effect/ change in the outcome before running these analyses.	Thank you for these considerations. The use of bootstrapping and other more nuanced details of the analysis will be shared in the separate data analysis plan, which will be shared via the Open Science Framework (OSF)	NA
9. 5. Ethics and dissemination – I would also mention here preregistration of the study protocol in Prospero, in line with guidelines for health-focussed systematic reviews and meta-	Thank you for spotting this. We have added this clarification as suggested “This IPD meta-analysis protocol is pre-registered (PROSPERO: CRD42022359691).”	p. 12, line 17

analyses (if not mentioned elsewhere in main text).		
10. Might also be worth clarifying if the resulting homogenised dataset for this meta-analysis will be made open access (to support future updates to the review/ meta-analysis in line with Cochrane recommendations)	Thank you for this comment. We have added a data availability statement under section 8. Unfortunately, we won't be able to make the dataset open access, since we are legally not allowed to share any data that we do not own and we are bound to strict data sharing contracts that we have with the original institutions specifying that the data can only be used for this project alone. Any data sharing requests will have to be made to the original institutions. “8. Data Availability Statement This IPD meta-analysis will use data that is obtained under data sharing contracts with the owners of individual data sets. Since these contracts do not allow for onward sharing, requests for IPD should be made to the original owners of the data.”	p. 16, lines 42 to 48

References

1. Stewart LA, Riley RD, Tierney JF. Planning and Initiating and IPD Meta-Analysis Project. In: Riley RD, Tierney JF, Stewart LA, eds. *Individual Participant Data Meta-Analysis: A Handbook for Healthcare Research*. First Edition. John Wiley & Sons Ltd.; 2021:22-24.
2. Riley RD, Stewart LA, Tierney JF. *Individual Participant Data Meta-Analysis for Healthcare Research*. Wiley; 2021. doi:10.1002/9781119333784.ch1
3. Tierney JF, Riley RD, Rydzewska LHM, Stewart LA. Running an IPD-Meta-Analysis Project. In: Riley RD, Tierney JF, Stewart LA, eds. *Individual Participant Data Meta-Analysis: A Handbook for Healthcare Research*. First Edition. John Wiley & Sons Ltd.; 2021:45-80.

VERSION 2 – REVIEW

REVIEWER	Hinton, Elanor University of Bristol
REVIEW RETURNED	27-Nov-2023

GENERAL COMMENTS	I have read the response letter and revised manuscript. All my comments have been adequately addressed and recommend this manuscript for publication. Good luck with the analysis - I'll look forward to reading the findings in due course.
--

REVIEWER	Embling, Rochelle Swansea University, School of Psychology
REVIEW RETURNED	13-Nov-2023

GENERAL COMMENTS	Thank you for the author response - I believe all comments have been sufficiently addressed.
--

VERSION 2 – AUTHOR RESPONSE